# Fast Vision Transformers with HiLo Attention

**Zizheng Pan**    **Jianfei Cai**    **Bohan Zhuang**[†]

Department of Data Science & AI, Monash University, Australia

## Abstract

Vision Transformers (ViTs) have triggered the most recent and significant break-throughs in computer vision. Their efficient designs are mostly guided by the indirect metric of computational complexity, *i.e.*, FLOPs, which however has a clear gap with the direct metric such as throughput. Thus, we propose to use the direct speed evaluation on the target platform as the design principle for efficient ViTs. Particularly, we introduce LITv2, a simple and effective ViT which performs favourably against the existing state-of-the-art methods across a spectrum of different model sizes with faster speed. At the core of LITv2 is a novel self-attention mechanism, which we dub **HiLo**. HiLo is inspired by the insight that high frequencies in an image capture local fine details and low frequencies focus on global structures, whereas a multi-head self-attention layer neglects the characteristic of different frequencies. Therefore, we propose to disentangle the high/low frequency patterns in an attention layer by separating the heads into two groups, where one group encodes high frequencies via self-attention within each local window, and another group encodes low frequencies by performing global attention between the average-pooled low-frequency keys and values from each window and each query position in the input feature map. Benefiting from the efficient design for both groups, we show that HiLo is superior to the existing attention mechanisms by comprehensively benchmarking FLOPs, speed and memory consumption on GPUs and CPUs. For example, HiLo is $1.4\times$ faster than spatial reduction attention and $1.6\times$ faster than local window attention on CPUs. Powered by HiLo, LITv2 serves as a strong backbone for mainstream vision tasks including image classification, dense detection and segmentation. Code is available at https://github.com/ziplab/LITv2.

## 1 Introduction

Real-world applications usually require a model to have an optimal speed and accuracy trade-off under limited computational budget, such as UAV and autonomous driving. This motivates substantial works toward efficient vision Transformer (ViT) design, such as PVT [51], Swin [32] and Focal Transformer [60], among others. To measure the computational complexity, a widely adopted metric in recent ViT design is the number of float-point operations, *i.e.*, FLOPs. However, FLOPs is an indirect metric, which can not directly reflect the real speed on the target platform. For example, Focal-Tiny is much slower than Swin-Ti on GPUs although their FLOPs are comparable.

In general, the discrepancy between the indirect metric (FLOPs) and the direct metric (speed) in recent ViTs can be attributed to two main reasons. First, although self-attention is efficient on low-resolution feature maps, the quadratic complexity in both memory and time makes it much slower on high-resolution images due to intensive memory access cost [34], where fetching data from off-chip DRAM can be speed-consuming. Second, some efficient attention mechanisms in ViTs have low theoretical complexity guarantee but are actually slow on GPUs due to particular operations that

---

[†]Corresponding author. E-mail: bohan.zhuang@monash.edu

36th Conference on Neural Information Processing Systems (NeurIPS 2022).

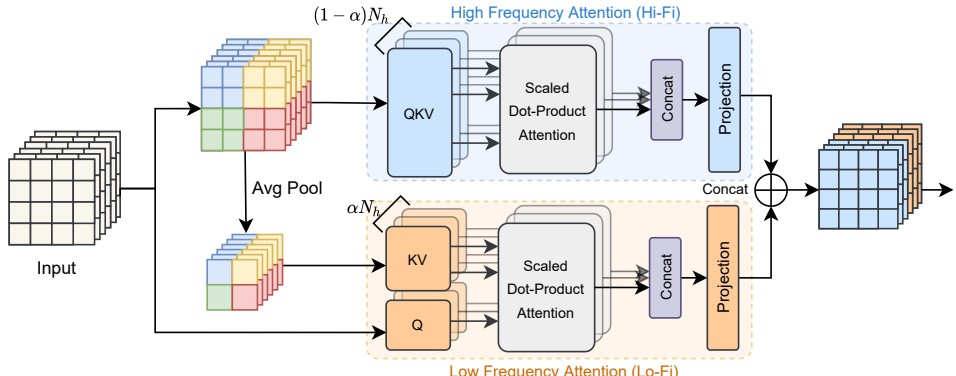

Figure 1: Framework of HiLo attention. $N_h$ refers to the total number of self-attention heads at this layer. $\alpha$ denotes the split ratio for high/low frequency heads. Best viewed in color.

are not hardware-friendly or cannot be parallelized, such as the multi-scale window partition [60], recursion [44] and dilated window [20].

With these observations, in this paper we propose to evaluate ViT by the direct metric, *i.e.*, throughput, not only FLOPs. Based on this principle, we introduce LITv2, a novel efficient and accurate vision Transformer that outperforms most state-of-the-art (SoTA) ViTs on standard benchmarks while being practically faster on GPUs. LITv2 is bulit upon LITv1 [36], a simple ViT baseline which removes all multi-head self-attention layers (MSAs) in the early stages while applying standard MSAs in the later stages. Benefit from this design, LITv1 is faster than many existing works on ImageNet classification due to no computational cost from the early MSAs while the later MSAs only need to process downsampled low-resolution feature maps. However, the standard MSA still suffers from huge computational cost on high-resolution images, especially for dense prediction tasks.

To address this problem, we propose a novel efficient attention mechanism, termed **HiLo**. HiLo is motivated by the fact that natural images contain rich frequencies where high/low frequencies play different roles in encoding image patterns, *i.e.*, local fine details and global structures, respectively. A typical MSA layer enforces the same global attention across all image patches without considering the characteristics of different underlying frequencies. This motivates us to propose to separate an MSA layer into two paths where one path encodes high-frequency interactions via local self-attention with relatively high-resolution feature maps while the other path encodes low-frequency interactions via global attention with down-sampled feature maps, which leads to a great efficiency improvement.

Specifically, HiLo employs two efficient attentions to disentangle **Hi**gh/**Lo**w frequencies in feature maps. As shown in Figure 1, in the upper path, we allocate a few heads to the high frequency attention (Hi-Fi) to capture fine-grained high frequencies by local window self-attention (*e.g.*, $2 \times 2$ windows), which is much more efficient than standard MSAs. The lower path, implementing the low-frequency attention (Lo-Fi), first applies average pooling to each window to obtain low-frequency signals. Then, we allocate the remaining heads for Lo-Fi to model the relationship between each query position in the input feature map and the average-pooled low-frequency keys and values from each window. Benefit from the reduced length of keys and values, Lo-Fi also achieves significant complexity reduction. Finally, we concatenate the refined high/low-frequency features and forward the resulting output into subsequent layers. Since both Hi-Fi and Lo-Fi are not equipped with time-consuming operations such as dilated windows and recursion, the overall framework of HiLo is fast on both CPUs and GPUs. We show by comprehensive benchmarks that HiLo achieves advantage over the existing attention mechanisms in terms of performance, FLOPs, throughput and memory consumption.

Besides, we find the fixed relative positional encoding in LITv1 dramatically slows down its speed on dense prediction tasks due to the interpolation for different image resolutions. For better efficiency, we propose to adopt one $3 \times 3$ depthwise convolutional layer with zero-padding in each FFN to incorporate the implicitly learned position information from zero-padding [27]. Moreover, the $3 \times 3$ convolutional filters simultaneously help to enlarge the receptive field of the early multi-layer perceptron (MLP) blocks in LITv1. Finally, we conduct extensive experiments on ImageNet, COCO and ADE20K to evaluate the performance of LITv2. Comprehensive comparisons with SoTA models show that our architecture achieves competitive performance with faster throughput, making ViTs more feasible to run low-latency applications for real-world scenarios.

## 2 Related Work

**Vision Transformers.** Vision Transformers are neural networks that adopt self-attention mechanisms into computer vision tasks. In [18], Dosovitskiy *et al.* propose a ViT for image classification, which inherits the similar architecture from a standard Transformer [48] in natural language processing (NLP) tasks. Since then, subsequent works have been proposed to improve ViT by incorporating more convolutional layers [54, 61], introducing pyramid feature maps [51, 32], enhancing the locality [62], as well as automatically searching a well-performed architecture [5, 3] with neural architecture search (NAS). Some others also seek for token pruning to accelerate the inference speed of ViTs [37] or applying ViT into low-level vision tasks [47]. Compared to existing works, this paper focuses on a general ViT-based backbone for computer vision (CV) tasks and aims to achieve better efficiency on GPUs while maintaining competitive performance.

**Efficient attention mechanisms.** Efficient attention mechanisms aim to reduce the quadratic complexity of standard MSAs. Existing efforts in NLP can be roughly categories into low-rank decomposition [50], kernelization [28, 39], memory [40] and sparsity mechanism [10]. However, simply adopting these method usually performs suboptimally in CV tasks [32, 63]. In CV, representative efficient self-attention mechanisms includes spatial reduction attention (SRA) [51], local window attention [32, 26] and Twins attention [12]. However, they only focus on either local or global attention at the same layer. To address this problem, TNT [21] introduced additional global tokens and MixFormer [6] mixed local window attention with depthwise convolutional layers. Some other attention mechanisms consider both simultaneously, such as Focal [60] and QuadTree [44]. However, due to the inefficient operations which are not hardware-friendly and cannot be reflected in FLOPs (*e.g.*, multi-scale window partition, recursion), they are slow on GPUs even compared to standard MSA. To this end, the proposed HiLo attention simultaneously captures rich local-global information at the same MSA layer and is faster and more memory-efficient compared to the existing works.

**Frequency domain analysis in vision.** The frequency domain analysis in CV has been well studied in the literature. According to [13, 16], the low frequencies in an image usually capture global structures and color information while the high frequencies contain fine details of objects (*e.g.*, sharp edges). Based on this insight, a plethora of solutions have been proposed for image super-resolution [66, 19], generalization [25], image re-scaling [56] and neural network compression [59, 7]. Furthermore, Octave convolution [9] targeted convolutional layers and proposed to locally applies convolution on high/low-resolution feature maps, separately. Different from it, the proposed HiLo is a novel attention mechanism that captures both local and global relationships with self-attention.

## 3 Background

**Multi-head self-attention.** Transformers are built upon multi-head self-attention, which enables to capture long-range relationships for tokens at different positions. Specifically, let $\mathbf{X} \in \mathbb{R}^{N \times D}$ be the input sequence into a standard MSA layer, where $N$ is the length of the input sequence and $D$ refers to the number of hidden dimensions. Each self-attention head calculates the query $\mathbf{Q}$, key $\mathbf{K}$ and value $\mathbf{V}$ matrices with a linear transformation from $\mathbf{X}$,

$$\mathbf{Q} = \mathbf{X}\mathbf{W}_q, \mathbf{K} = \mathbf{X}\mathbf{W}_k, \mathbf{V} = \mathbf{X}\mathbf{W}_v, \tag{1}$$

where $\mathbf{W}_q, \mathbf{W}_k, \mathbf{W}_v \in \mathbb{R}^{D \times D_h}$ are learnable parameters and $D_h$ is the number of hidden dimensions for a head. Next, the output of a self-attention head is a weighted sum over $N$ value vectors,

$$\mathrm{SA}_h(\mathbf{X}) = \mathrm{Softmax}(\frac{\mathbf{Q}\mathbf{K}^\top}{\sqrt{D_h}})\mathbf{V}. \tag{2}$$

For an MSA layer with $N_h$ heads, the final output is computed by a linear projection of the concatenated outputs from each self-attention head, which can be formulated by

$$\mathrm{MSA}(\mathbf{X}) = \underset{h \in [N_h]}{\mathrm{concat}}[\mathrm{SA}_h(\mathbf{X})]\mathbf{W}_o, \tag{3}$$

where $\mathbf{W}_o \in \mathbb{R}^{(N_h \times D_h) \times D}$ is a learnable parameter. In practice, $D$ is usually equal to $N_h \times D_h$. Overall, a standard MSA layer have the computational cost of $4ND^2 + 2N^2D$, where $2N^2D$ comes from Eq. (2), $3ND^2$ and $ND^2$ comes from Eq. (1) and Eq. (3), respectively.

**Transformer blocks.** A standard vision Transformer as described in [18] consists of a patch embedding layer, several blocks and a prediction head. Let $l$ be the index of a block. Then each block contains an MSA layer and a position-wise feed-forward network (FFN), which can expressed as

$$\mathbf{X}'_{l-1} = \mathbf{X}_{l-1} + \mathrm{MSA}(\mathrm{LN}(\mathbf{X}_{l-1})), \tag{4}$$

$$\mathbf{X}_l = \mathbf{X}'_{l-1} + \mathrm{FFN}(\mathrm{LN}(\mathbf{X}'_{l-1})), \tag{5}$$

where LN denotes the LayerNorm [2] and an FFN consists of two FC layers with GELU [24] non-linearity in between. Recent works on ViT have proposed to divide the blocks into several stages (typically 4 stages) to generate pyramid feature maps for dense prediction tasks. Furthermore, to reduce the computational cost on high-resolution feature maps in the early stages, the MSA in Eq. (4) has been replaced with efficient alternatives, such as SRA [51] and W-MSA [32].

**Bottlenecks of LITv1.** Recent studies have shown that the MSA layers in the early stages in a model still focus on local patterns [14]. With the same observation, LITv1 [36] removes all early MSAs (*i.e.*, exclude Eq. (4) in each block) while applying standard MSAs at the later stages. This design principle has achieved better efficiency with competitive performance on ImageNet compared to PVT [51] and Swin [32]. However, LITv1 still has two main bottlenecks in speed: 1) Given a high-resolution image, the standard MSAs in the later stages still result in huge computational cost. 2) The fixed relative positional encoding [32] dramatically slows down the speed when dealing with different image resolutions. This is due to interpolating the fixed-size positional encoding for each different image resolution. In the next section, we describe a novel attention mechanism with zero padding positional encoding to comprehensively accelerate LITv1.

## 4 Method

### 4.1 HiLo Attention

We propose to separately process high/low frequencies in a feature map at an attention layer. We name the new attention mechanism as HiLo, which is depicted in Figure 1. Essentially, the low-frequency attention branch (Lo-Fi) is to capture the global dependencies of the input (image/features), which does not need a high-resolution feature map but requires global attention. On the other hand, the high-frequency attention branch (Hi-Fi) is to capture the fine detailed local dependency, which requires a high-resolution feature map but can be done via local attention. In the next, we describe the two attentions in detail.

**High-frequency attention.** Intuitively, as high frequencies encode local details of objects, it can be redundant and computationally expensive to apply global attention on a feature map. Therefore, we propose to design Hi-Fi to capture fine-grained high frequencies with local window self-attention (*e.g.*, $2 \times 2$ windows), which saves significant computational complexity. Furthermore, we employ the simple non-overlapping window partition in Hi-Fi, which is more hardware-friendly compared to the time-consuming operations such as window shifting [32] or multi-scale window partition [60].

**Low-frequency attention.** Recent studies have shown that the global attention in MSA helps to capture low frequencies [38]. However, directly applying MSA to high-resolution feature maps requires huge computational cost. As averaging is a low-pass filter [49], Lo-Fi firstly applies average pooling to each window to get low-frequency signals in the input $\mathbf{X}$. Next, the average-pooled feature maps are projected into keys $\mathbf{K} \in \mathbb{R}^{N/s^2 \times D_h}$ and values $\mathbf{V} \in \mathbb{R}^{N/s^2 \times D_h}$, where $s$ is the window size. The queries $\mathbf{Q}$ in Lo-Fi still comes from the original feature map $\mathbf{X}$. We then apply the standard attention to capture the rich low-frequency information in feature maps. Note that due to the spatial reduction of $\mathbf{K}$ and $\mathbf{V}$, Lo-Fi simultaneously reduces the complexity for both Eq. (1) and Eq. (2).

**Head splitting.** A naive solution for head assignment is to allocate both Hi-Fi and Lo-Fi the same number of heads as the standard MSA layer. However, doubling heads results in more computational cost. In order to achieve better efficiency, HiLo separates the same number of heads in an MSA into two groups with a split ratio $\alpha$, where $(1-\alpha)N_h$ heads will be employed for Hi-Fi and the other $\alpha N_h$ heads are used for Lo-Fi. By doing so, as each attention has a lower complexity than a standard MSA, the entire framework of HiLo guarantees a low complexity and ensures high throughput on GPUs. Moreover, another benefit of head splitting is that the learnable parameter $\mathbf{W}_o$ can be decomposed into two smaller matrices, which helps to reduce model parameters. Finally, the output of HiLo is a

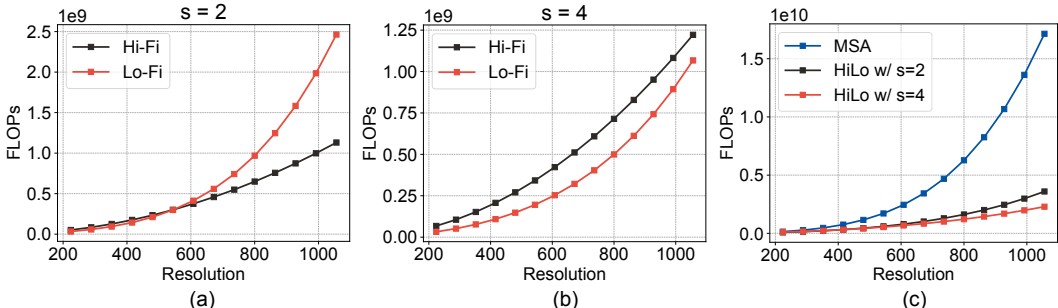

Figure 2: FLOPs comparison for Hi-Fi and Lo-Fi under different image resolutions and equal number of heads (Figures a and b). A larger window size helps HiLo achieve better efficiency on high-resolution images (Figure c).

concatenation of the outputs from each attention

$$\text{HiLo}(\mathbf{X}) = [\text{Hi-Fi}(\mathbf{X}); \text{Lo-Fi}(\mathbf{X})], \tag{6}$$

where $[\cdot]$ denotes the concatenation operation.

**Complexity Analysis.** Without loss of generality, we assume Hi-Fi and Lo-Fi have an equal number of heads (*i.e.*, $\alpha = 0.5$) and the feature map has equal width and height. Then, Hi-Fi and Lo-Fi have a computational cost of $\frac{7}{4}ND^2 + s^2ND$ and $(\frac{3}{4} + \frac{1}{s^2})ND^2 + \frac{1}{s^2}N^2D$, respectively. Derivation for this result can be found in the supplementary material. As shown in Figure 2-(a) and (b), under a small input image resolution and a small value of $s$ (*e.g.*, $s = 2$), both Hi-Fi and Lo-Fi are comparably efficient. However, with a much higher resolution, Lo-Fi will result in a huge computational cost as it still has a quadratic complexity in terms of $N$ in Eq. (2), *i.e.*, $\frac{1}{s^2}N^2D$. In this case, slightly increasing $s$ (*e.g.*, $s = 4$) helps Lo-Fi achieve better efficiency while preserving the accuracy. Combining the two attentions together, a larger window size also helps the overall framework of HiLo to reduce more FLOPs on high-resolution images, as shown in Figure 2-(c). Thus, we suggest a practical guideline for adopting HiLo into existing frameworks: *increasing the window size in order to get better efficiency on high-resolution images*. We further show in Section 5.2 that this principle helps LITv2 achieve a better speed and accuracy trade-off on downstream tasks, *e.g.*, dense object detection.

### 4.2 Positional Encoding

Positional encoding is essential to self-attention due to its permutation-invariant property. In LITv1, the later MSAs adopt the same relative positional encoding (RPE) scheme as Swin [32]. This approach has significantly improves Swin by 0.7% in Top-1 accuracy on ImageNet compared to using absolute positional encoding [32]. However, on dense prediction tasks, the fixed RPE has to be interpolated for different image resolutions, which dramatically slows down the training/inference speed of LITv1. As a recent study [27] has shown that position information can be implicitly learned from zero-padding in CNNs, we propose to adopt one layer of $3 \times 3$ depthwise convolutional layer with zero-padding in each FFN to replace the time-consuming RPE. Notably, due to the elimination of early MSAs, the early blocks in LITv1 only have FFNs left, which results in a tiny receptive field of $1 \times 1$. To this end, we show in Section 5.4 that the $3 \times 3$ convolutional filters adopted in each FFN also improve LITv2 by simultaneously enlarging the receptive field in the early stages.

### 4.3 Model Architecture

LITv2 has three variants: LITv2-S, LITv2-M and LITv2-B, corresponding to the small, medium and base settings in LITv1, respectively. For a fair comparison, we keep the network width and depth as the same as LITv1. The overall modifications are simply in two steps: 1) Adding one layer of depthwise convolution with zero-padding in each FFN and removing all relative positional encodings in all MSAs. 2) Replacing all attention layers with the proposed HiLo attention. Detailed architecture configurations can be found in the supplementary material.

# 5 Experiment

In this section we conduct experiments to validate the effectiveness of the proposed LITv2. Following common practice [51, 32, 12, 60], we experiment LITv2 on three tasks, including image classification on ImageNet-1K [43], object detection and instance segmentation on COCO [31] and semantic segmentation on ADE20K [65].

Table 1: Image classification results on ImageNet-1K. By default, the FLOPs, throughput and memory consumption are measured based on the resolution $224 \times 224$. We report the throughput and training/test time memory consumption with a batch size of 64. Throughput is tested on one NVIDIA RTX 3090 GPU and averaged over 30 runs. ResNet results are from "ResNet Stikes Back" [53]. "↑ 384" means a model is finetuned at the resolution $384 \times 384$. "OOM" means "out-of-memory".

| Model | Param (M) | FLOPs (G) | Throughput (imgs/s) | Train Mem (GB) | Test Mem (GB) | Top-1 (%) |
|---|---|---|---|---|---|---|
| ResNet-50 [53] | 26 | 4.1 | 1,279 | 7.9 | 2.8 | 80.4 |
| ConvNext-Ti [33] | 28 | 4.5 | 1,079 | 8.3 | 1.7 | 82.1 |
| PVT-S [51] | 25 | 3.8 | 1,007 | 6.8 | 1.3 | 79.8 |
| Swin-Ti [32] | 28 | 4.5 | 961 | 6.1 | 1.5 | 81.3 |
| CvT-13 [54] | 20 | 4.5 | 947 | 6.1 | 1.5 | 81.6 |
| Focal-Tiny [60] | 29 | 4.9 | 384 | 12.2 | 3.3 | **82.2** |
| Twins-PCPVT-S [12] | 24 | 3.8 | 998 | 6.8 | 1.2 | 81.2 |
| LITv1-S [36] | 27 | 4.1 | 1,298 | 5.8 | 1.2 | 81.5 |
| **LITv2-S** | 28 | 3.7 | **1,471** | **5.1** | **1.2** | 82.0 |
| ResNet-101 [53] | 45 | 7.9 | 722 | 10.5 | 3.0 | 81.5 |
| ConvNext-S [33] | 50 | 8.7 | 639 | 12.3 | 1.8 | 83.1 |
| PVT-M [51] | 44 | 6.7 | 680 | 9.3 | 1.5 | 81.2 |
| Twins-SVT-B [12] | 56 | 8.3 | 621 | 9.8 | 1.9 | 83.2 |
| Swin-S [32] | 50 | 8.7 | 582 | 9.7 | 1.7 | 83.0 |
| LITv1-M [36] | 48 | 8.6 | 638 | 12.0 | 1.4 | 83.0 |
| **LITv2-M** | 49 | 7.5 | **812** | **8.8** | **1.4** | **83.3** |
| ResNet-152 [53] | 60 | 11.6 | 512 | 13.4 | 2.9 | 82.0 |
| ConvNext-B [33] | 89 | 15.4 | 469 | 16.9 | 2.9 | **83.8** |
| Twins-SVT-L [12] | 99 | 14.8 | 440 | 13.7 | 3.1 | 83.7 |
| Swin-B [32] | 88 | 15.4 | 386 | 13.4 | 2.4 | 83.3 |
| LITv1-B [36] | 86 | 15.0 | 444 | 16.4 | 2.1 | 83.4 |
| **LITv2-B** | 87 | 13.2 | **602** | **12.2** | **2.1** | 83.6 |
| DeiT-B↑ 384 [45] | 86 | 55.4 | 159 | 39.9 | **2.5** | 83.1 |
| Swin-B↑ 384 [32] | 88 | 47.1 | 142 | OOM | 6.1 | 84.5 |
| **LITv2-B↑ 384** | 87 | 39.7 | **198** | **35.8** | 4.6 | **84.7** |

## 5.1 Image Classification on ImageNet-1K

We conduct image classification experiments on ImageNet-1K [43], a large-scale image dataset which contains ∼1.2M training images and 50K validation images from 1K categories. We measure the model performance by Top-1 accuracy. Furthermore, we report the FLOPs, throughput, as well as training/test memory consumption on GPUs. We compare with two CNN-based models [53, 33] and several representative SoTA ViTs [51, 32, 54, 60, 12]. Note that this paper does not consider mobile-level architectures [8, 35]. Instead, we focus on models with the similar model size. Besides, we are also not directly comparable with NAS-based methods [3, 5] as LITv2 is manually designed.

**Implementation details.** All models are trained for 300 epochs from scratch on 8 V100 GPUs. At training time, we set the total batch size as 1,024. The input images are resized and randomly cropped into $224 \times 224$. The initial learning rate is set to $1 \times 10^{-3}$ and the weight decay is set to $5 \times 10^{-2}$. We use AdamW optimizer with a cosine decay learning rate scheduler. All training strategies including the data augmentation are same as in LITv1. For HiLo, the window size $s$ is set to 2. The split ratio $\alpha$ is set to 0.9, which is chosen from a simple grid search on ImageNet-1K. The depthwise convolutional layers in FFNs are set with a kernel size of $3 \times 3$, stride of 1 and zero padding size of 1.

Table 2: Object detection and instance segmentation performance on the COCO `val2017` split using the RetinaNet [30] and Mask R-CNN [22] framework. $AP^b$ and $AP^m$ denote the bounding box AP and mask AP, respectively. "*" indicates the model adopts a local window size of 4 in HiLo.

| Backbone | RetinaNet | | | | Mask R-CNN | | | | |
|---|---|---|---|---|---|---|---|---|---|
| | Params | FLOPs (G) | FPS | $AP^b$ | Params | FLOPs (G) | FPS | $AP^b$ | $AP^m$ |
| ResNet-50 [23] | 38M | 239 | 18.5 | 36.3 | 44M | 260 | 27.1 | 38.0 | 34.4 |
| PVT-S [51] | 34M | 273 | 13.0 | 40.4 | 44M | 292 | 16.2 | 40.4 | 37.8 |
| Swin-T [32] | 38M | 251 | 17.0 | 41.5 | 48M | 270 | 21.1 | 42.2 | 39.1 |
| Twins-SVT-S [12] | 34M | 225 | 15.5 | 43.0 | 44M | 244 | 20.4 | 43.4 | 40.3 |
| LITv1-S [36] | 39M | 305 | 3.3 | 41.6 | 48M | 324 | 3.2 | 42.9 | 39.6 |
| **LITv2-S** | 38M | 242 | 18.7 | **44.0** | 47M | 261 | 18.7 | **44.9** | **40.8** |
| **LITv2-S\*** | 38M | 230 | 20.4 | 43.7 | 47M | 249 | 21.9 | 44.7 | 40.7 |
| ResNet-101 [23] | 57M | 315 | 15.2 | 38.5 | 63M | 336 | 20.9 | 40.4 | 36.4 |
| PVT-M [51] | 54M | 348 | 10.5 | 41.9 | 64M | 367 | 10.8 | 42.0 | 39.0 |
| Swin-S [32] | 60M | 343 | 13.3 | 44.5 | 69M | 362 | 15.8 | 44.8 | 40.9 |
| Twins-SVT-B [12] | 67M | 358 | 10.8 | 45.3 | 76M | 377 | 12.7 | 45.2 | 41.5 |
| **LITv2-M** | 59M | 348 | 12.2 | **46.0** | 68M | 367 | 12.6 | **46.8** | **42.3** |
| **LITv2-M\*** | 59M | 312 | 14.8 | 45.8 | 68M | 315 | 16.0 | 46.5 | 42.0 |
| ResNeXt101-64x4d [58] | 96M | 473 | 10.3 | 41.0 | 102M | 493 | 12.4 | 42.8 | 38.4 |
| PVT-L [51] | 71M | 439 | 9.5 | 42.6 | 81M | 457 | 8.3 | 42.9 | 39.5 |
| Swin-B [32] | 98M | 488 | 11.0 | 44.7 | 107M | 507 | 11.3 | 45.5 | 41.3 |
| Twins-SVT-L [12] | 111M | 504 | 9.9 | 45.7 | 120M | 524 | 10.1 | 45.9 | 41.6 |
| **LITv2-B** | 97M | 481 | 9.5 | **46.7** | 106M | 500 | 9.3 | **47.3** | **42.6** |
| **LITv2-B\*** | 97M | 430 | 11.8 | 46.3 | 106M | 449 | 11.5 | 46.8 | 42.3 |

**Results.** In Table 1, we report the experiment results on ImageNet-1K. First, compared to LITv1 baselines, LITv2 achieves consistent improvement on Top-1 accuracy while using less FLOPs. Moreover, benefit from HiLo, LITv2 achieves faster throughput and significant training time memory reduction (*e.g.*, 13%, 27%, 36% inference speedup for the small, medium and base settings, respectively) compared to LITv1. Second, compared to CNNs, LITv2 models outperform all counterparts of ResNet and ConvNext in terms of FLOPs, throughput and memory consumption while achieving comparable performance. Last, compared to SoTA ViTs, LITv2 surpasses many models in terms of throughput and memory consumption with competitive performance. For example, under the similar amount of FLOPs, LITv2-S achieves faster inference speed than PVT-S and Twins-PCPVT-S with better performance. Although Focal-Tiny achieves better Top-1 accuracy than LITv2-S, it runs much slower (*i.e.*, 384 vs. 1,471 images/s) and requires a large amount of memory to train. Besides, when finetuning on a higher resolution, LITv2-B outperforms both DeiT-B and Swin-B with a faster throughput and lower complexity.

## 5.2 Object Detection and Instance Segmentation on COCO

In this section, we conduct experiments on COCO 2017, a common benchmark for object detection and instance segmentation which contains ∼118K images for the training set and ∼5K images for the validation set. Following common practice [12, 51], we experiment with two detection frameworks: RetinaNet [30] and Mask R-CNN [22]. We measure model performance by Average Precision (AP).

**Implementation details.** All backbones are initialized with pretrained weights on ImageNet-1K. We train each model on 8 GPUs with $1\times$ schedule (12 epochs) and a total batch size of 16. For a fair comparison, we adopt the same training strategy and hyperparameter settings as in LITv1 [36]. Note that we pretrain LITv2 with a local window size of 2 and $\alpha = 0.9$ on ImageNet-1K. Under the same $\alpha$, a larger window size helps to achieve lower complexity and thus improves the speed at high resolution, as explained in Section 4.1. In this case, we also train models with a slightly larger window size of $s = 4$ for better efficiency, which we denote with "*". By default, FLOPs is evaluated based on the input resolution of $1280 \times 800$. FPS is measured on one RTX 3090 GPU based on the mmdetection [4] framework.

**Results.** In Table 2, we report the experimental results on COCO. In general, LITv2 outperforms LITv1 by a large margin in almost all metrics. Besides, our LITv2 significantly surpasses ResNet in terms of AP, though it runs slightly slower in some cases. More importantly, our LITv2 beats all the compared SoTA ViTs, achieving the best AP with compelling fast inference speed. Furthermore, by adopting a larger window size (*i.e.*, $s = 4$), LITv2 achieves better efficiency with a slightly performance drop.

Table 4: Performance comparisons with other efficient attention mechanisms in ViTs based on LITv2-S. We report the Top-1 accuracy on ImageNet-1K and mIoU on ADE20K.

| Method | ImageNet-1K | | | | | | ADE20K | | |
|---|---|---|---|---|---|---|---|---|---|
| | Params (M) | FLOPs (G) | Throughput (images/s) | Train Mem (GB) | Test Mem (GB) | Top-1 (%) | Params (M) | FLOPs (G) | mIoU (%) |
| MSA | 28 | 4.1 | 1,293 | 6.5 | 1.2 | 82.3 | 32 | 46.5 | 43.7 |
| SRA [51] | 32 | 4.0 | 1,425 | 5.1 | 1.3 | 81.7 | 35 | **42.4** | 42.8 |
| W-MSA [32] | 28 | 4.0 | 1,394 | 5.3 | 1.2 | 81.9 | 32 | 42.7 | 41.9 |
| T-MSA [12] | 30 | 4.0 | 1,462 | **5.0** | 1.3 | 81.8 | 33 | 42.5 | 44.0 |
| HiLo | **28** | **3.7** | **1,471** | 5.1 | **1.2** | 82.0 | **31** | 42.6 | **44.3** |

Figure 3: Comparison with other attention mechanisms based on LITv2-S. We report the FLOPs, throughput, and training/test time memory consumption. Evaluations are based on a batch size of 64 on one RTX 3090 GPU. The black cross symbol means "out-of-memory".

## 5.3 Semantic Segmentation on ADE20K

In this section, we evaluate LITv2 on the semantic segmentation task. We conduct experiments on ADE20K [65], a widely adopted dataset for semantic segmentation which has ∼20K training images, ∼2K validation images and ∼3K test images. Following prior works, we adopt the framework of Semantic FPN [29] and measure the model performance by mIoU. We train each model on 8 GPUs with a total batch size of 16 with 80K iterations. All backbones are initialized with pretrained weights on ImageNet-1K. The stochastic depth for the small, medium and base models of LITv2 are 0.2, 0.2 and 0.3, respectively. All other training strategies are the same as in LITv1 [36].

**Results.** In Table 3, we compare LITv2 with ResNet and representative ViTs on ADE20K. In general, LITv2 achieves fast speed while outperforming many SoTA models. For example, our LITv2-S, LITv2-M and LITv2-B surpass Swin-Ti, Swin-S and Swin-B by 2.8%, 0.5% and 1.2% in mIoU with higher FPS, respectively.

Table 3: Semantic segmentation performance of different backbones on the ADE20K validation set. FLOPs is evaluated based on the image resolution of $512 \times 512$.

| Backbone | Params (M) | FLOPs (G) | FPS | mIoU (%) |
|---|---|---|---|---|
| ResNet-50 [23] | 29 | 45 | 45.4 | 36.7 |
| PVT-S [51] | 28 | 40 | 38.7 | 39.8 |
| Swin-Ti [32] | 32 | 46 | 39.6 | 41.5 |
| Twins-SVT-S [12] | 28 | 37 | 34.5 | 43.2 |
| LITv1-S [36] | 32 | 46 | 18.1 | 41.7 |
| **LITv2-S** | 31 | 41 | 42.6 | **44.3** |
| ResNet-101 [23] | 48 | 66 | 36.7 | 38.8 |
| PVT-M [51] | 48 | 55 | 29.7 | 41.6 |
| Swin-S [32] | 53 | 70 | 24.4 | 45.2 |
| Twins-SVT-B [12] | 60 | 67 | 28.0 | 45.3 |
| **LITv2-M** | 52 | 63 | 28.5 | **45.7** |
| PVT-L [51] | 65 | 71 | 20.5 | 42.1 |
| Swin-B [32] | 107 | 107 | 25.5 | 46.0 |
| Twins-SVT-L [12] | 104 | 102 | 25.9 | 46.7 |
| **LITv2-B** | 90 | 93 | 27.5 | **47.2** |

## 5.4 Ablation Study

In this section, we provide ablation studies for LITv2, including the comparison with other efficient attention variants, the effect of $\alpha$ in HiLo, as well as the effect of architecture modifications. By default, the throughput and memory consumption are measured on one RTX 3090 GPU with a batch size of 64 under the resolution of $224 \times 224$.

**Comparing HiLo with other attention mechanisms.** Based on LITv2-S, we compare the performance of HiLo with other efficient attention mechanisms on ImageNet-1K, including spatial reduction attention (SRA) in PVT [51], shifted-window based attention (W-MSA) in Swin [32] and alternated local and global attention (T-MSA) in Twins [12]. In our implementation, we directly

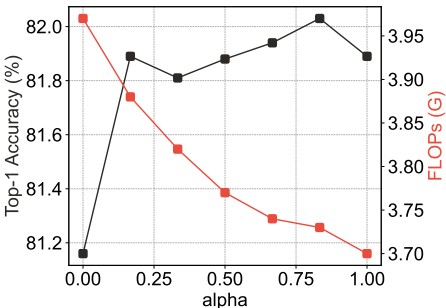

Figure 4: Effect of $\alpha$ based on LITv2-S.

Table 5: Effect of architecture modifications based on LITv1-S. "ConvFNN" means we add one layer of $3 \times 3$ depthwise convolutional layer into each FFN. "RPE" refers to relative positional encoding [32].

| Name | ImageNet-1K | | | COCO (RetinaNet) | | |
|---|---|---|---|---|---|---|
| | FLOPs (G) | Mem (GB) | Top-1 (%) | FLOPs (G) | FPS | AP |
| LITv1-S [36] | 4.1 | 5.8 | 81.5 | 305 | 3.3 | 41.6 |
| + ConvFFN | 4.1 | 6.5 | 82.5 | 306 | 3.1 | 45.1 |
| + Remove RPE | 4.1 | 6.5 | 82.3 | 306 | 13.3 | 44.7 |
| + HiLo | 3.7 | 5.1 | 82.0 | 224 | 18.7 | 44.0 |

Figure 5: Frequency magnitude ($14 \times 14$) from 8 output channels of Hi-Fi and Lo-Fi in LITv2-B. The magnitude is averaged over 100 samples. The lighter the color, the larger the magnitude. A pixel that is closer to the centre means a lower frequency. Visualization code can be found in the supplementary material.

replace HiLo with each compared method. The results are reported in Table 4. In general, HiLo reduces more FLOPs while achieving better performance and faster speed than the compared methods. Furthermore, in Figure 3, we provide comprehensive benchmarks for more attention mechanisms based on different image resolutions, including Focal [60], QuadTree [44] and Performer [11]. Suffering from weak parallelizability, they are even slower than that of using standard MSAs on GPUs. Compared to them, HiLo achieves competitive results in terms of the FLOPs, throughput and memory consumption. Moreover, we conduct experiments based on ADE20K and Semantic FPN and show that HiLo achieves more performance gain than other attention mechanisms on the downstream dense prediction task.

**Effect of $\alpha$.** As shown in Figure 4, since the complexity of Lo-Fi is lower than Hi-Fi under the resolution of $224 \times 224$ and the window size of 2, a larger $\alpha$ helps to reduce more FLOPs as we allocate more heads to Lo-Fi. Moreover, we found HiLo performs badly with $\alpha = 0$, in which case only the Hi-Fi is left and HiLo only focuses on high frequencies. We speculate that low frequencies play an important role in self-attention. For other values of $\alpha$, we find the performance difference is around 0.2%, where $\alpha = 0.9$ achieves the best performance. However, it is worth noting that although the pure Lo-Fi branch ($\alpha = 1.0$) can achieve competitive results on ImageNet-1K, high-frequency signals play an important role in capturing fine object details, which is particularly important for dense prediction tasks such as semantic segmentation. For example, with $\alpha = 0.9$, LITv2-S based Semantic FPN achieves more performance gain (+0.6%) than that of using $\alpha = 1.0$ (43.7%).

**Effect of architecture modifications.** Based on LITv2-S, we explore the effect of architecture modifications. As shown in Table 5, benefit from the enlarged receptive field in the early stages, the adoption of depthwise convolutions improves the performance on both ImageNet and COCO. Next, by removing the relative positional encoding, we significantly improve FPS on dense prediction tasks with a slightly performance drop on both datasets. Also note that since depthwise convolutions have encoded positional information by zero paddings [27], the elimination of RPE does not result in a significant performance drop compared to prior works [32]. Finally, benefit from HiLo, we achieve more gains in model efficiency on both ImageNet and COCO.

**Spectrum analysis of HiLo.** In Figure 5, we visualize the magnitude of frequency component [42] by applying Fast Fourier Transform (FFT) to the output feature maps from Hi-Fi and Lo-Fi attentions, respectively. The visualisation indicates that Hi-Fi captures more high frequencies and Lo-Fi mainly focuses on low frequencies. This strongly aligns with our aim of disentangling high and low frequencies in feature maps at a single attention layer.

Table 6: Speed and performance comparisons between LITv2-S and other recent ViTs on different GPUs. All throughput results are averaged over 30 runs with a total batch size of 64 and image resolution of $224 \times 224$ on one GPU card. We also report the Top-1 accuracy on ImageNet-1K.

| Model | Params (M) | FLOPs (G) | A100 | V100 | RTX 6000 | RTX 3090 | Top-1 (%) |
|---|---|---|---|---|---|---|---|
| ResNet-50 [53] | 26 | 4.1 | 1,424 | 1,123 | 877 | 1,279 | 80.4 |
| PVT-S [51] | 25 | 3.8 | 1,460 | 798 | 548 | 1,007 | 79.8 |
| Twins-PCPVT-S [12] | 24 | 3.8 | 1,455 | 792 | 529 | 998 | 81.2 |
| Swin-Ti [32] | 28 | 4.5 | 1,564 | 1,039 | 710 | 961 | 81.3 |
| TNT-S [21] | 24 | 5.2 | 802 | 431 | 298 | 534 | 81.3 |
| CvT-13 [54] | 20 | 4.5 | 1,595 | 716 | 379 | 947 | 81.6 |
| CoAtNet-0 [15] | 25 | 4.2 | 1,538 | 962 | 643 | 1,151 | 81.6 |
| CaiT-XS24 [46] | 27 | 5.4 | 991 | 484 | 299 | 623 | 81.8 |
| PVTv2-B2 [52] | 25 | 4.0 | 1,175 | 670 | 451 | 854 | 82.0 |
| XCiT-S12 [1] | 26 | 4.8 | 1,727 | 761 | 504 | 1,068 | 82.0 |
| ConvNext-Ti [33] | 28 | 4.5 | 1,654 | 762 | 571 | 1,079 | 82.1 |
| Focal-Tiny [60] | 29 | 4.9 | 471 | 372 | 261 | 384 | **82.2** |
| **LITv2-S** | 28 | **3.7** | **1,874** | **1,304** | **928** | **1,471** | 82.0 |

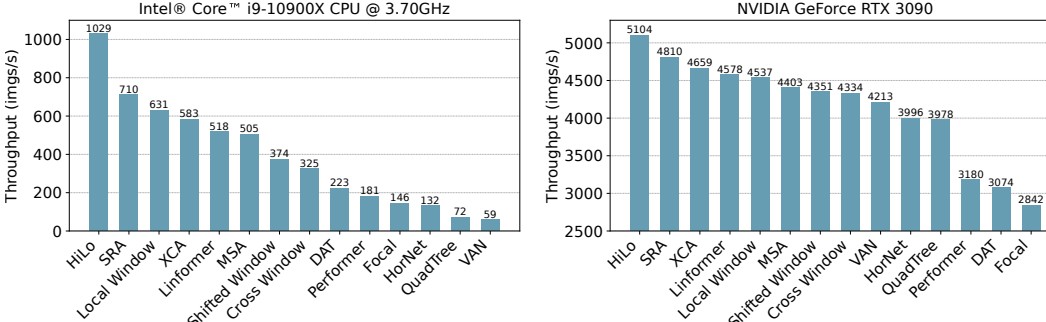

Figure 6: Throughput comparisons with more attention mechanisms on CPUs and GPUs based on a single attention layer and $14 \times 14$ feature maps.

**Speed and performance comparisons with more ViTs on different GPUs.** We compare the inference speed with more models and on more types of GPUs. Table 6 reports the results. It shows that LITv2-S still achieves consistent faster throughput (images/s) than many ViTs on NVIDIA A100, Tesla V100, RTX 6000, and RTX 3090. It is also worth noting that under similar performance (82.0%), LITv2-S is 2.1× faster than PVTv2-B2 [52], 1.7× faster than XCiT-S12 [1] and ConvNext-Ti [33], and 3.5× faster than Focal-Tiny [60] on V100, which is another common GPU version for speed test in previous works [32, 45, 33, 64].

**Throughput comparisons with more attention mechanisms on CPUs and GPUs.** In Figure 6, we show that HiLo is consistently faster than many attention mechanisms [51, 32, 1, 50, 11, 17, 55, 60, 41, 44, 20, 18] on both CPUs and GPUs. In particular, under CPU testing, HiLo is 1.4× faster than SRA [51], 1.6× faster than local window attention [32] and 17.4× faster than VAN [20]. Detailed benchmark configurations can be found in the supplementary material.

## 6 Conclusion and Future Work

In this paper, we have introduced LITv2, a novel efficient vision Transformer backbone with fast speed on GPUs and outperforms most SoTA models on ImageNet and downstream tasks. We have also presented HiLo attention, the core of LITv2 which helps to achieve better efficiency especially on high-resolution images. With competitive performance, HiLo achieves great advantage over the existing attention mechanisms across FLOPs, throughput and memory consumption. Future work may include incorporating convolutional stem [57] and overlapping patch embedding [52] for better performance, or extending HiLo on more tasks such as speech recognition and video processing.

**Limitations and societal impact.** HiLo adopts a head splitting ratio to assign different numbers of heads into Hi-Fi and Lo-Fi. In our experiments, this ratio is determined by a grid search on ImageNet (*i.e.*, $\alpha = 0.9$). However, different tasks may have different importance on high and low frequencies. Thus, the optimal value of $\alpha$ is task-specific and needs to be set manually. Besides, our work potentially brings some negative societal impacts, such as the huge energy consumption and carbon emissions from large-scale training on GPU clusters.

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
