# Supplementary Material for Fast Vision Transformers with HiLo Attention

**Zizheng Pan**     **Jianfei Cai**     **Bohan Zhuang**[*]

Department of Data Science & AI, Monash University, Australia

We organize our supplementary material as follows.

- In Section A, we describe the architecture specifications of LITv2.
- In Section B, we provide the derivation for the computational cost of HiLo attention.
- In Section C, we study the effect of window size based on CIFAR-100.
- In Section D, we provide additional study on the Lo-Fi branch where we directly compute the queries from pooled feature maps.
- In Section E, we describe more details of the throughput benchmark for more attention mechanisms on CPUs and GPUs.
- In Section F, we provide more visualisation examples for spectrum analysis of HiLo attention.

Table I: Architecture specifications of LITv2. $P$ denotes the patch size in the patch embedding layer and $C$ is the channel dimension. $H$ is the number of self-attention heads. $\alpha$ and $s$ are the split ratio and window size in HiLo, respectively. $E$ is the expansion ratio in the FFN layer. "DTM" refers to the deformable token merging module in LITv1. We use "ConvFFN Block" to differentiate our modified FFNs in the early stages from the previous MLP Blocks in LITv1 [8].

| Stage | Output Size | Layer Name | LITv2-S | LITv2-M | LITv2-B |
|---|---|---|---|---|---|
| Stage 1 | $\frac{H}{4} \times \frac{W}{4}$ | Patch Embedding | $P_1 = 4$ 
 $C_1 = 96$ | $P_1 = 4$ 
 $C_1 = 96$ | $P_1 = 4$ 
 $C_1 = 128$ |
| | | ConvFFN Block | $\begin{bmatrix} E_1 = 4 \end{bmatrix} \times 2$ | $\begin{bmatrix} E_1 = 4 \end{bmatrix} \times 2$ | $\begin{bmatrix} E_1 = 4 \end{bmatrix} \times 2$ |
| Stage 2 | $\frac{H}{8} \times \frac{W}{8}$ | DTM | $P_2 = 2$ 
 $C_2 = 192$ | $P_2 = 2$ 
 $C_2 = 192$ | $P_2 = 2$ 
 $C_2 = 256$ |
| | | ConvFFN Block | $\begin{bmatrix} E_2 = 4 \end{bmatrix} \times 2$ | $\begin{bmatrix} E_2 = 4 \end{bmatrix} \times 2$ | $\begin{bmatrix} E_2 = 4 \end{bmatrix} \times 2$ |
| Stage 3 | $\frac{H}{16} \times \frac{W}{16}$ | DTM | $P_3 = 2$ 
 $C_3 = 384$ | $P_3 = 2$ 
 $C_3 = 384$ | $P_3 = 2$ 
 $C_3 = 512$ |
| | | Transformer Block | $\begin{bmatrix} \alpha_3 = 0.9 \\ s_3 = 2 \\ H_3 = 12 \\ E_3 = 4 \end{bmatrix} \times 6$ | $\begin{bmatrix} \alpha_3 = 0.9 \\ s_3 = 2 \\ H_3 = 12 \\ E_3 = 4 \end{bmatrix} \times 18$ | $\begin{bmatrix} \alpha_3 = 0.9 \\ s_3 = 2 \\ H_3 = 16 \\ E_3 = 4 \end{bmatrix} \times 18$ |
| Stage 4 | $\frac{H}{32} \times \frac{W}{32}$ | DTM | $P_4 = 2$ 
 $C_4 = 768$ | $P_4 = 2$ 
 $C_4 = 768$ | $P_4 = 2$ 
 $C_4 = 1024$ |
| | | Transformer Block | $\begin{bmatrix} \alpha_4 = 1.0 \\ s_4 = 1 \\ H_4 = 24 \\ E_4 = 4 \end{bmatrix} \times 2$ | $\begin{bmatrix} \alpha_4 = 1.0 \\ s_4 = 1 \\ H_4 = 24 \\ E_4 = 4 \end{bmatrix} \times 2$ | $\begin{bmatrix} \alpha_4 = 1.0 \\ s_4 = 1 \\ H_4 = 32 \\ E_4 = 4 \end{bmatrix} \times 2$ |

[*]Corresponding author. E-mail: bohan.zhuang@monash.edu

36th Conference on Neural Information Processing Systems (NeurIPS 2022).

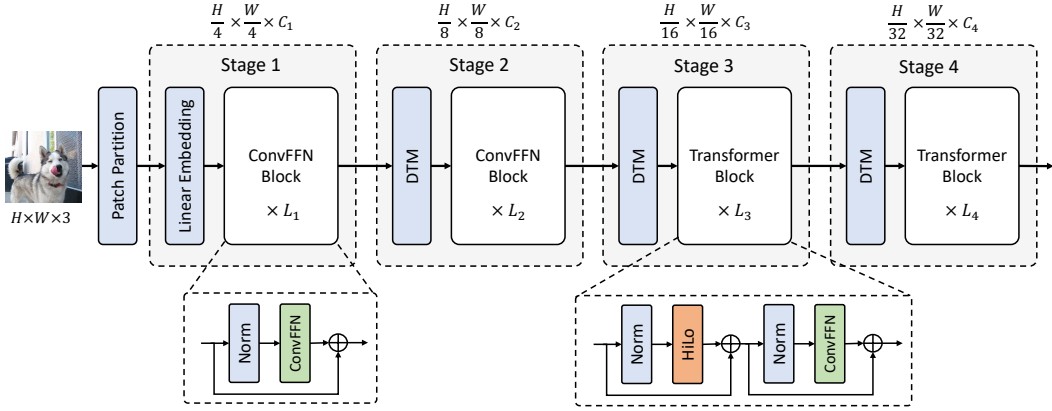

Figure I: Framework of LITv2. $C_i$ and $L_i$ refer to the number of hidden dimensions and the number of blocks at the $i$-th stage. "ConvFFN" denotes our modified FFN layer where we adopt one layer of depthwise convolution in the FFN.

## A    Architecture Specifications of LITv2

The overall framework of LITv2 is depicted in Figure I. We also provide detailed architecture specifications of LITv2 in Table I. In general, we set the same network depth and width as LITv1. It is worth noting that recent works [12, 7, 3, 15, 13] usually adopt standard MSAs at the last stage, including LITv1. Following common practice, we set $\alpha = 1.0$ and $s = 1$ at the last stage to make HiLo behave as a standard MSA. LITv2 also excludes MSAs in the first two stages due to the tiny receptive field of attention heads, as visualized in Figure 3 of LITv1 [8].

## B    Computational Cost of HiLo Attention

Let $N$ and $D$ be the number of tokens and the number of hidden dimensions in an HiLo attention layer. We denote $s$ as the window size. For simplicity, we assume Hi-Fi and Lo-Fi have an equal number of heads and the feature map has equal width and height. Then, the computational cost of each attention comes from three parts: 1) The projections of $\mathbf{Q}$, $\mathbf{K}$, $\mathbf{V}$ matrices. 2) The attention computation and weighted-sum of values. 3) The final linear projection of the weighted-sum values. For Hi-Fi, the computational cost for each part is

$$N \times D \times \frac{D}{2} \times 3 = \frac{3}{2}ND^2, \tag{1}$$

$$s^2 \times s^2 \times \frac{D}{2} \times \frac{N}{s^2} \times 2 = s^2ND, \tag{2}$$

$$N \times \frac{D}{2} \times \frac{D}{2} = \frac{1}{4}ND^2, \tag{3}$$

respectively. Overall, this gives rise to a total computational cost of $\frac{7}{4}ND^2 + s^2ND$ for Hi-Fi. Next, the computational cost for each part in Lo-Fi is

$$N \times D \times \frac{D}{2} + \frac{N}{s^2} \times D \times \frac{D}{2} \times 2 = (\frac{1}{2} + \frac{1}{s^2})ND^2, \tag{4}$$

$$N \times \frac{N}{s^2} \times \frac{D}{2} \times 2 = \frac{N^2}{s^2}D, \tag{5}$$

$$N \times \frac{D}{2} \times \frac{D}{2} = \frac{1}{4}ND^2, \tag{6}$$

respectively. Thus, the total computational cost of Lo-Fi is $(\frac{3}{4} + \frac{1}{s^2})ND^2 + \frac{1}{s^2}N^2D$.

Table II: Effect of window size based on LITv2-S. We report the Top-1 accuracy on CIFAR-100.

| Window Size | Params (M) | FLOPs (G) | Throughput (imgs/s) | Train Memory (GB) | Test Memory (GB) | Top-1 (%) |
|---|---|---|---|---|---|---|
| 2 | **27** | 3.7 | **1,476** | 5.1 | **1.2** | **85.1** |
| 3 | 27 | 3.7 | 1,437 | 5.1 | 1.2 | 84.6 |
| 4 | 27 | 3.8 | 1,417 | 5.1 | 1.2 | 84.4 |
| 5 | 27 | 3.7 | 1,434 | 5.1 | 1.2 | 84.6 |
| 6 | 27 | 3.9 | 1,413 | 5.2 | 1.2 | 84.8 |
| 7 | 27 | **3.6** | 1,442 | **4.9** | 1.2 | 84.8 |

Table III: Effect of directly computing queries from the pooled feature maps. We report the Top-1 accuracy on ImageNet-1K.

| Model | Params (M) | FLOPs (G) | Throughput (imgs/s) | Top-1 (%) |
|---|---|---|---|---|
| LITv2-S | 28 | 3.7 | **1,471** | **82.0** |
| LITv2-S w/ pooled queries | 28 | 3.5 | 1,084 | 81.9 |

## C  Effect of Window Size

Based on LITv2-S, we study the effect of window size in HiLo by experimenting on CIFAR-100. As shown in Table II, the window size does not affect the model parameters since the parameters of HiLo do not depend on it. Moreover, as both Hi-Fi and Lo-Fi are comparably efficient under the small resolution of image classification (*i.e.*, 224×224), all settings have the comparable FLOPs, speed and memory footprint, where the difference is mainly due to the extra cost from padding on feature maps for window partition [7]. Overall, we find the window size of 2 performs the best, which therefore serves as our default setting in LITv2 for image classification. Also note that as discussed in the main manuscript, a slightly larger window size (*e.g.*, 4) can help LITv2 achieve better efficiency on larger resolutions with a slightly performance drop.

## D  Additional Study on the Lo-Fi Branch

In the proposed HiLo attention, the Lo-Fi branch computes queries from the original input feature maps. An alternative approach is to directly compute the queries from the average-pooled feature maps. However, in self-attention, the number of queries determines the spatial size of the output feature maps. When computing the queries from pooled feature maps, the spatial size of the output feature maps is inconsistent with that of the original input. One solution is to use interpolation (e.g. bilinear) and concatenate the interpolated feature maps with the outputs from Hi-Fi. However, as shown in Table III, this approach (denoted as "pooled queries") brings inferior performance and much slower throughput than our proposed design. Note that although computing queries from pooled feature maps can slightly achieve a lower theoretical model complexity, frequently applying interpolation on GPUs results in a high memory access cost (MAC). Therefore, it instead slows down the inference speed on GPUs.

## E  Details of Throughput Benchmark for Different Attention Mechanisms

To evaluate the inference speed of HiLo on CPUs and GPUs, we benchmark the throughput based on a single attention layer and the standard settings of training ViT-B [5] on ImageNet. Specifically, under the input resolution of 224×224, attention layers in ViT-B need to handle 14×14 (1/16 scale) feature maps, where each attention layer has 12 heads and each head has 64 dimensions. For a fair comparison, we adopt the aforementioned configurations for all compared methods by default. Besides, since different methods have distinct hyperparameters, we adopt their default settings for dealing with 1/16 scale feature maps. For example, HiLo adopts a window size of 2 and alpha of 0.9 when processing 1/16 scale feature maps. In Table IV, we report more benchmark results. Overall,

Table IV: Throughput benchmark for different attention mechanisms based on a single attention layer. We report the throughput on both CPU (Intel® Core™ i9-10900X CPU @ 3.70GHz) and GPU (NVIDIA GeForce RTX 3090).

| Name | Params (M) | FLOPs (M) | CPU (imgs/s) | GPU (imgs/s) |
|------|-----------|-----------|--------------|--------------|
| MSA [5] | 2.36 | 521.4 | 505 | 4,403 |
| Cross Window [4] | 2.37 | 493.3 | 325 | 4,334 |
| DAT [14] | 2.38 | 528.7 | 223 | 3,074 |
| Performer [2] | 2.36 | 617.2 | 181 | 3,180 |
| Linformer [11] | 2.46 | 616.6 | 518 | 4,578 |
| SRA [12] | 4.72 | 419.6 | 710 | 4,810 |
| Local Window [7] | 2.36 | 477.2 | 631 | 4,537 |
| Shifted Window [7] | 2.36 | 477.2 | 374 | 4,351 |
| Focal [15] | 2.44 | 526.9 | 146 | 2,842 |
| XCA [1] | 2.36 | 481.7 | 583 | 4,659 |
| QuadTree [10] | 5.33 | 613.3 | 72 | 3,978 |
| VAN [6] | 1.83 | 358.0 | 59 | 4,213 |
| HorNet [9] | 2.23 | 436.5 | 132 | 3,996 |
| HiLo | 2.20 | **298.3** | **1,029** | **5,104** |

we show that under a similar amount of parameters, a single layer of HiLo uses less FLOPs than compared methods, meanwhile it is faster on both CPUs and GPUs.

## F  More Visualisations on Spectrum Analysis

In Figure II and Figure III, we provide frequency magnitude visualisations for Hi-Fi and Lo-Fi attention outputs, respectively. Clearly, the results indicate that Hi-Fi captures more high frequencies in LITv2 while Lo-Fi mainly focuses on low frequencies. We also provide the PyTorch-style code in Algorithm 1 to explain our visualisation.

---

**Algorithm 1** PyTorch-style Code for Visualising Frequency Magnitude.

---

```python
import matplotlib.pyplot as plt
import torch

def visualize_freq(x):
    '''
    x : The output feature maps from either Hi-Fi or Lo-Fi attention.
        Tensor shape: (batch_size, hidden_dim, height, width)
    '''
    fft_output = torch.fft.fft2(x.float())
    freq_img = torch.log(torch.abs(torch.fft.fftshift(fft_output)))
    num_plots = 8

    # average over samples
    freq_img_mean = freq_img.mean(dim=0).cpu()
    fig, axis = plt.subplots(1, num_plots, figsize=(num_plots * 4, 4))

    for i in range(num_plots):
        axis[i].imshow(freq_img_mean[i, ...].numpy())
        axis[i].axes.xaxis.set_visible(False)
        axis[i].axes.yaxis.set_visible(False)

    plt.axis('off')
    plt.tight_layout()
    plt.show()
```

---

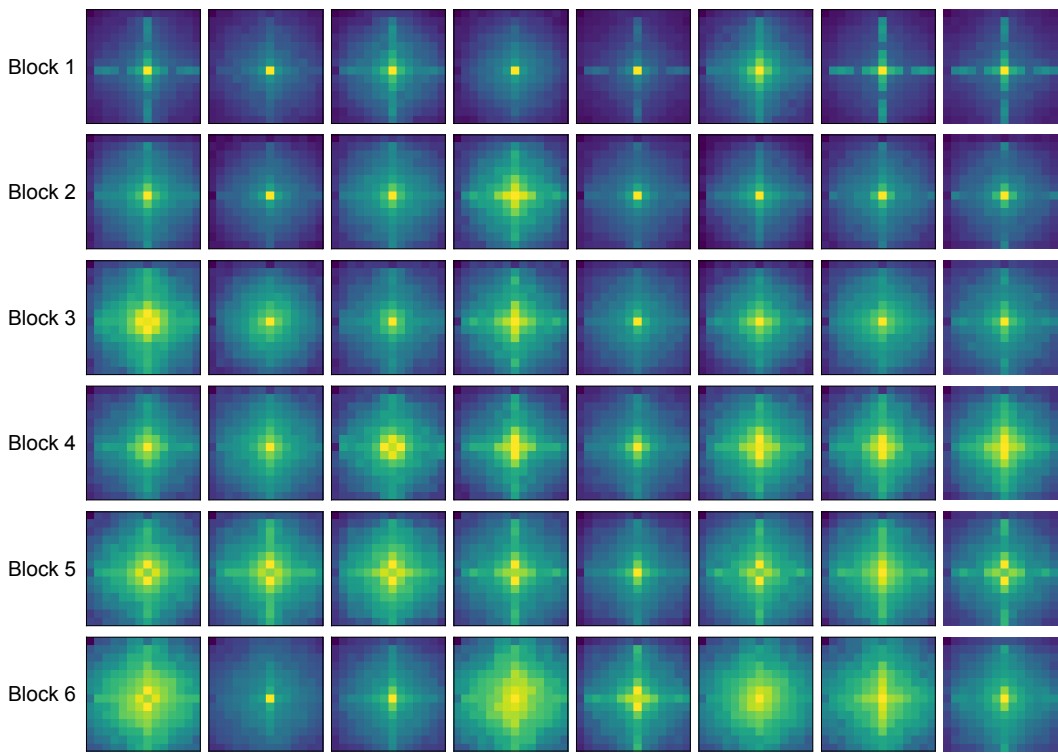

Figure II: Frequency magnitude ($14 \times 14$) from 8 output channels of Hi-Fi in LITv2-S. The magnitude is averaged over 100 samples.

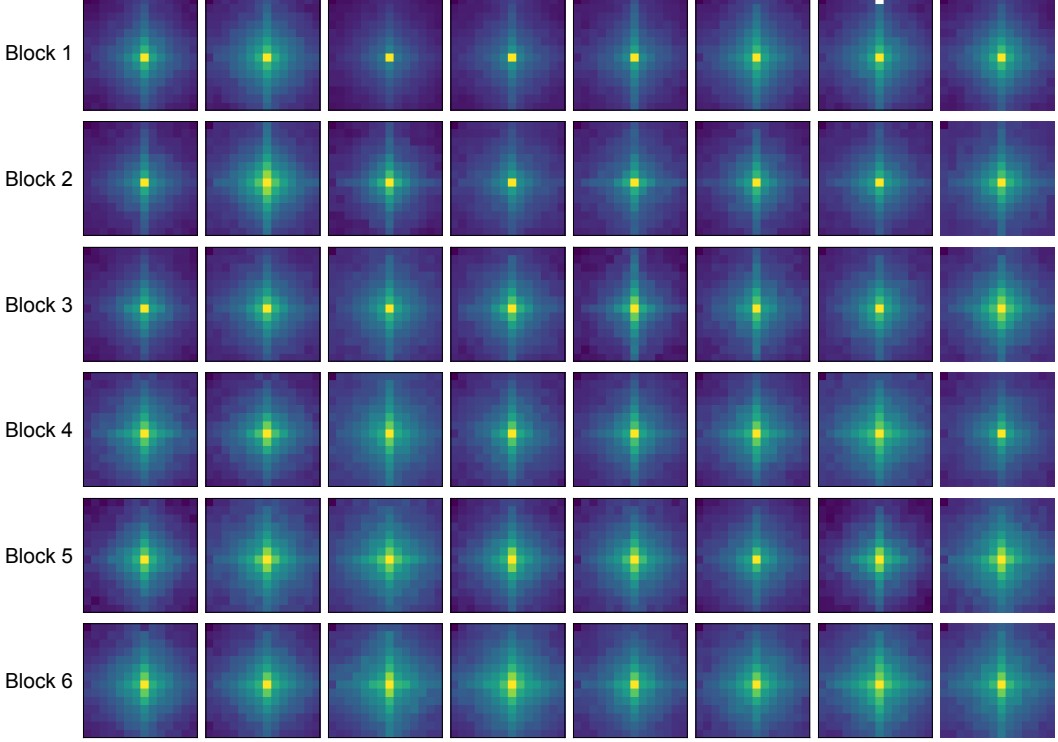

Figure III: Frequency magnitude ($14 \times 14$) from 8 output channels of Lo-Fi in LITv2-S. The magnitude is averaged over 100 samples.