# OpenReview forum: "Fast Vision Transformers with HiLo Attention"
_NeurIPS.cc/2022/Conference — NeurIPS 2022 Accept_

### Official Review · Reviewer_Rjxy · 2022-07-08

**Rating:** 6
**Confidence:** 5
**Soundness:** 3 good
**Presentation:** 3 good
**Contribution:** 3 good

**Summary:**

This work introduces HiLo Attention to vision transformers on top of LIT[1] for 2D images. The proposed HiLo is composed of High-frequency attention (Hi-Fi) and Low-frequency attention (Lo-Fi). By splitting two branches in every transformer block and combining the output of the two branches, these two attention modules process the high-frequency information and low-frequency information in the image. This work achieves competitive results in Image Classification on ImageNet-1K, Object Detection and Instance Segmentation on COCO, and Semantic Segmentation on ADE20K. The work also shows the visualization of low-frequency features and high-frequency features.




[1] Pan, Zizheng, et al. "Less is more: Pay less attention in vision transformers." Proceedings of the AAAI Conference on Artificial Intelligence. Vol. 36. No. 2. 2022.

**Questions:**

1. What is the strict definition of low-frequency/high-frequency attention? How do you define the relationship between high/low attentions with local/global attentions

2. How do you generate the figure 5 ? (visualization of high/low feature)

3. [Minor] How do you compare your method and results with C-SWin[1]?




[1] Dong, Xiaoyi, et al. "Cswin transformer: A general vision transformer backbone with cross-shaped windows." Proceedings of the IEEE/CVF Conference on Computer Vision and Pattern Recognition. 2022.


**Strengths And Weaknesses:**

Strengths:

1. This work addresses the problem of modeling local and global features in vision transformers, which is good motivation from the perspective of Fourier transform.

2. This work achieves state-of-the-art performance on different tasks e.g. Image Classification on ImageNet-1K, Object Detection and Instance Segmentation on COCO, and Semantic Segmentation on ADE20K.

3. Many implementation details are mentioned in the paper e.g. positional encoding, and low-FLOP models not always being fast models

Weaknesses:

1. There is little discussion or proof to show the connection between low/high frequency and local/global features. The visualization of low/high frequency feature (fig 5) is confusing for reading to understand.

2. Looking at the Figure 4, the best model with alpha=0.8 is only roughly higher than the model with alpha=1.0. In the case of alpha=1.0, there is no global attention (or low-frequency feature) is involved. This ablation study looks very confusing. Does it mean global attention (or low-frequency attention) is useless?

3. [Minor concern] Although comparing FLOPs is not a fair speed comparison, the throughput of a certain kind of GPU (RTX 3090) doesn't lead to a strong conclusion.

---

> ### Author Response · Authors · 2022-08-02
> **Response to Reviewer Rjxy - Part 1**
>
>
> We thank you for your valuable feedback and address your questions as follows.
>
> **Q1. Connection between low/high frequency and local/global features. Fig. 5 is confusing.**
>
> Essentially, the low-frequency attention branch (Lo-Fi) is to capture the global dependencies of the input (image/features), which does not need a high-resolution feature map but requires global attention. On the other hand, the high-frequency attention branch (Hi-Fi) is to capture the fine detailed local dependency, which requires a high-resolution feature map but can be done via local attention. Our idea at a high level is generally similar to the low/high frequency concepts in the classic digital image processing, where low-frequency components that captures global structure tend to have long-range correlations while high-frequency components that captures local sharp changes such as edges tend to have more short-range correlations. We have added more discussions in Section 4.1 in the revision.
>
> Fig. 5 is obtained by simply applying Fast Fourier Transform (FFT) on both the Hi-Fi branch output and the Lo-Fi branch output and visualising the magnitudes of their frequency components. As stated in Line 314 (Line 307 of the initial submission), from Fig. 5 we can see that the Hi-Fi output contains more high frequencies (local features) while the Lo-Fi output contains more low frequencies (global features). We have added the visualisation code for Fig. 5 in the revised supplementary file.
>
>
> **Q2. Is global attention (or low-frequency attention) useless?**
>
> We believe the reviewer misunderstood the results in Fig. 4. To clarify, alpha = 1.0 means only the Lo-Fi branch is left (Lines 165-167, or Lines 163-164 in the initial submission). Thus, we assume the question should be "Is the local attention or high-frequency attention useless?". To answer this question, we conducted an experiment with alpha=1.0 on ADE20K. We show that comparing with alpha=0.9, simply using Lo-Fi results in more performance drop (0.6%), as shown in our response to Reviewer zE7q Q2. This experiment demonstrates that both high frequencies and low frequencies are essential in CV tasks. In particular, we speculate that image classification mainly focuses on the global information of the entire image, and thus low frequencies perform favourably as they capture global structure. However, dense prediction tasks usually require fine object details, for which high frequencies play an important role.

---

> ### Author Response · Authors · 2022-08-02
> **Response to Reviewer Rjxy - Part 2**
>
>
> **Q3. The throughput of a certain kind of GPU (RTX 3090) doesn't lead to a strong conclusion.**
>
> Thanks for your advice. Following the same settings of Table 1, we compare the inference speed with more models and on more types of GPUs. The table below reports the results. It shows that LITv2-S still achieves consistent faster throughput (images/s) than many ViTs on NVIDIA A100, Tesla V100, RTX 6000, and RTX 3090. It is also worth noting that under similar performance (~82.0%), LITv2-S is $2.1\times$ faster than PVTv2-B2, $1.7\times$ faster than XCiT-S12 and ConvNext-Ti, and $3.5\times$ faster than Focal-Tiny on V100, which is another common GPU version for speed test in previous works [30,29,41] ([25,26,36] in the initial submission). We have added this comparison into the revised supplementary file.
>
> | Model         | Params (M) | FLOPs (G) | A100      | V100      | RTX 6000 | RTX 3090  | Top-1 (%) |
> | ------------- | ---------- | --------- | --------- | --------- | -------- | --------- | --------- |
> | ResNet-50     | 26         | 4.1       | 1,424     | 1,123     | 877      | 1,279     | 80.4      |
> | PVT-S         | 25         | 3.8       | 1,460     | 798       | 548      | 1,007     | 79.8      |
> | Twins-PCPVT-S | 24         | 3.8       | 1,455     | 792       | 529      | 998       | 81.2      |
> | Swin-Ti       | 28         | 4.5       | 1,564     | 1,039     | 710      | 961       | 81.3      |
> | TNT-S         | 24         | 5.2       | 802       | 431       | 298      | 534       | 81.3      |
> | CvT-13        | 20         | 4.5       | 1,595     | 716       | 379      | 947       | 81.6      |
> | CoAtNet-0     | 25         | 4.2       | 1,538     | 962       | 643      | 1,151     | 81.6      |
> | CaiT-XS24     | 27         | 5.4       | 991       | 484       | 299      | 623       | 81.8      |
> | PVTv2-B2      | 25         | 4.0       | 1,175     | 670       | 451      | 854       | 82.0      |
> | XCiT-S12      | 26         | 4.8       | 1,727     | 761       | 504      | 1,068     | 82.0      |
> | ConvNext-Ti   | 28         | 4.5       | 1,654     | 762       | 571      | 1,079     | 82.1      |
> | Focal-Tiny    | 29         | 4.9       | 471       | 372       | 261      | 384       | **82.2**  |
> | LITv2-S       | 28         | **3.7**   | **1,874** | **1,304** | **928**  | **1,471** | 82.0      |
>
>
> **Q5. Compared with CSwin.**
>
> CSwin Transformer proposes a Cross-Shaped Window Self-Attention (CSwin), which divides the feature maps by horizontal and vertical stripes and uses self-attention to capture local dependencies. Compared to it, the proposed HiLo-based LITv2 is much easier to train and more scalable in terms of both the throughput and the training time memory consumption on GPUs.
>
> For example, the table below compares the training time memory consumption (MB) under different input image resolutions with a total batch size of 64 on one RTX 3090. "OOM" means out of memory.
>
> | Model   | Params (M) | FLOPs (G) | 224x224   | 256x256   | 288x288   | 320*320    | 352x352   | 384x384    |
> | ------- | ---------- | --------- | --------- | --------- | --------- | ---------- | --------- | ---------- |
> | CSwin-T | 23         | 4.3       | 9,224     | 13,207    | 16,375    | OOM        | OOM       | OOM        |
> | LITv2-S | 28         | 3.7       | **5,211** | **6,671** | **8,390** | **10,350** | **12564** | **15,045** |
>
> The table below compares the throughput (images/s) under different input image resolutions with a total batch size of 64 on one RTX 3090. Results are averaged over 30 runs.
>
> | Model   | Params (M) | FLOPs (G) | 224x224   | 256x256   | 288x288 | 320*320 | 352x352 | 384x384 |
> | ------- | ---------- | --------- | --------- | --------- | ------- | ------- | ------- | ------- |
> | CSwin-T | 23         | 4.3       | 814       | 591       | 481     | 376     | 297     | 262     |
> | LITv2-S | 28         | 3.7       | **1,471** | **1,128** | **895** | **718** | **591** | **493** |
>
> Also note that CSwin adopts a different training strategy compared with common practice [45,29,11] ([40,25,9] in the initial submission), e.g. more training epochs (310 v.s. 300 epochs) and larger batch size training (2,048 v.s. 1,024), as shown in their official released code. Therefore, directly comparing the model performance with CSwin can be inappropriate.

---

> ### Author Response · Authors · 2022-08-07
> **Request for discussion**
>
> Dear Reviewer Rjxy,
>
> We sincerely thank you again for your great efforts in reviewing this paper. We have addressed your main concerns regarding more discussions on the connection between high/low frequency and local/global features in the proposed HiLo, as well as the further explanation of Figure 5 and the importance of high-frequency attention on ADE20K. Please don’t hesitate to let us know if there are still some concerns/questions.
>
> Best regards,
>
> Authors of #310

---

### Official Review · Reviewer_VgYF · 2022-07-10

**Rating:** 9
**Confidence:** 5
**Soundness:** 4 excellent
**Presentation:** 4 excellent
**Contribution:** 4 excellent

**Summary:**

The paper proposes a novel vision Transformer, LITv2, which directly targets the speed on GPUs, instead of theoretical FLOPs. To achieve this, the authors improve the efficiency of ViT by targeting the architecture design, attention mechanism and positional encoding. Specifically, they first adopt the same design principle of LIT [29] to get rid of the early MSAs, then propose a novel efficient self-attention mechanism, HiLo, which disentangles the high/low frequency patterns in an attention layer with two groups of heads, where one group captures high frequencies via local window self-attention and another group of heads model the global relationship between the average-pooled low-frequency keys from each window and each query position in the input feature map. Comprehensive benchmarking has shown the advantage of the proposed HiLo over other attention schemes. Furthermore, the authors also propose to replace the time-consuming relative positional encoding (RPE) with zero-padding positional information from convolutions for further speedup on dense prediction tasks. Extensive experiments on both ImageNet and downstream tasks have shown that LITv2 achieves a better speed-accuracy trade-off than previous SoTA ViTs.

**Questions:**

See the weakness.

**Limitations:**

Yes.

**Strengths And Weaknesses:**

### Strength

1. The problem that the paper tackles for the current ViT design is significant. As most recent works focus only on theoretical FLOPs, the direct speed metric (e.g. throughput) on hardware is more important and informative for the community, especially for the efficient design of self-attention.

2. The proposed HiLo attention is well-motivated. The idea of disentangling different frequencies is technically sound and has been well validated by visualisations. The complexity analysis of HiLo is clear and the comparison with other efficient attention mechanisms is also comprehensive.

3. Simple architecture but works. The proposed model is backed by impressive results on both ImageNet and downstream tasks. The experiments are quite strong and the comparisons with other methods are comprehensive. Surprisingly, the proposed model achieves faster speed and uses less memory footprint than representative CNNs, with competitive performance on ImageNet. This is a significant step for general ViT backbones.

4. The structure of this paper is well-written and easy to follow. Nice figures, which clearly present the idea and claims.

### Weakness

1. The authors have indicated that by training a slightly larger window size the model can achieve better efficiency. However, it could be difficult to determine the optimal window size on dense prediction tasks. How would the speed-accuracy trade-off change if directly evaluating a trained model (e.g. trained with window size 2) with a different window size?

2. Swin Transformer also adopts RPE, but it is not slow on GPUs. So why removing RPE can significantly improve the speed of LITv1 on COCO?

3. It would be better if the COCO results based on LITv2-B can be moved into the main manuscript.

---

> ### Author Response · Authors · 2022-08-02
> **Response to Reviewer VgYF**
>
>
> Thanks for your very positive comments! We address your questions as follows.
>
> **Q1. Directly evaluating a trained model (e.g. trained with window size 2) with a different window size?**
>
> We agree that the window size still needs to be manually determined at the current stage. In this case, future work may consider automatically searching for a better window size by NAS. Moreover, we directly test the trained model of LITv2-S with RetinaNet under different window sizes. In this setting, the model is pretrained on COCO with a window size of 2. As shown in the table below, directly evaluating our LITv2-S based RetinaNet with different window sizes does not significantly hurt the performance. Instead, it brings different speed and accuracy trade-offs. This implies that in practice, one can set different window sizes for different speed requirements under a single trained model.
>
> | Window Size | FLOPs (G) | FPS  | AP   | AP_50 | AP_75 | AP_S | AP_M | AP_L |
> | ----------- | --------- | ---- | ---- | ----- | ----- | ---- | ---- | ---- |
> | 2           | 242       | 18.2 | 44.0 | 65.2  | 47.1  | 27.2 | 48.1 | 58.0 |
> | 3           | 234       | 19.6 | 43.7 | 64.7  | 47.1  | 27.0 | 47.5 | 58.0 |
> | 4           | 230       | 19.8 | 43.4 | 64.3  | 46.5  | 26.6 | 47.1 | 57.9 |
> | 5           | 228       | 20.0 | 43.0 | 63.9  | 46.0  | 27.0 | 46.7 | 57.3 |
> | 6           | 229       | 20.3 | 42.6 | 63.4  | 45.2  | 26.0 | 46.3 | 56.5 |
> | 7           | 229       | 20.4 | 42.4 | 63.0  | 45.1  | 25.9 | 46.1 | 56.4 |
>
>
>
> **Q2. Why removing RPE can significantly improve the speed of LITv1 on COCO.**
>
> The main reason is that Swin uses fixed-size local windows (e.g. 7$\times$7). Thus it does not need to interpolate the parameters of the fixed-size relative positional embedding. However, the global self-attention in LITv1 requires frequently interpolating the fixed-size relative positional embedding to adapt different sizes of the attention maps, as indicated in Lines 135-137 ( 132-134 in the initial submission). Therefore, removing RPE can significantly improve the speed of LITv1 on dense prediction tasks.
>
> **Q3. LITv2-B COCO results can be moved into the main manuscript.**
>
> Thanks for the suggestion. We have moved this result into Table 2 in the revised main manuscript.

---

> > ### Comment · Reviewer_VgYF · 2022-08-08
> > **Post-rebuttal discussion**
> >
> > Thank the authors for the response.
> >
> > Given the clear novelty of the HiLo attention and the strong results of the proposed architecture, I initially voted to accept this paper. After the rebuttal, I think the authors have addressed my main concerns. The speed-accuracy trade-off of window size on COCO object detection is reasonable (not significantly hurting the performance under different window sizes). The explanation of the difference in RPE between Swin and LITv1 is also evident.
> >
> > I have also carefully read the comments from other reviewers and the corresponding author responses:
> >
> > - Reviewer zE7q mainly questioned the importance of high-frequency attention in HiLo, for which the authors have provided additional results and shown more performance gain of Hi-Fi on ADE20K. The results on both ImageNet-1K and ADE20K have shown a clear advantage of HiLo over other attention mechanisms in ViTs.
> >
> > - Reviewer xTew mainly criticized the novelty and the additional comparisons with more ViTs, for which I think the main novelty of this paper is a new efficient co-design of local and global self-attention from a frequency perspective (disentangle frequency into low/high components and then customize redundancy draining respectively in a self-attention layer), as well as a fast and accurate SOTA ViT backbone for mainstream CV tasks. The research challenge of aligning the theoretical FLOPs with real speed is fundamental. In sharp contrast, existing attention mechanisms either separately process the local and global self-attention in ViTs or are much slower on GPUs than HiLo. For example, Focal and Quadtree have a clear gap between theoretical complexity and the direct speed metric (i.e. throughput). The advantage of the proposed architecture over other existing ViTs is more pronounced with additional comparisons (e.g. XCiT, CaiT).
> >
> > - Reviewer Rjxy is mainly concerned about the relation between high/low frequency and local/global features in HiLo as well as the visualization of Figure 5. From my point of view, the provided discussion is helpful and the explanation regarding Figure 5 is also clear.
> >
> > Overall, I think this paper is well-written, with a strong novelty and promising experimental results, solving a critical research gap. The revision has also reflected the changes. Therefore, I decided to keep my original rating.

---

> > > ### Author Response · Authors · 2022-08-08
> > > **Response to Post-rebuttal Discussion**
> > >
> > > Thanks for your very detailed reviews and for appreciating our work! Please feel free to ask any further questions.

---

### Official Review · Reviewer_xTew · 2022-07-10

**Rating:** 6
**Confidence:** 4
**Soundness:** 4 excellent
**Presentation:** 4 excellent
**Contribution:** 3 good

**Summary:**

The paper addresses efficient Vision Transformers (ViTs) design. The paper argues that while previous works on designing efficient ViTs have considered the theoretical asymptotic computational complexity and computational complexity measured in floating point operations (FLOPS) and memory, those metrics do not capture the actual running time and throughput. Specifically, the paper argues that previous methods might require low number of FLOPs (or lower asymptotic complexity), but in practise their implementation is not hardware friendly thus slow when running on GPU. The paper proposes to benchmark FLOPS, memory consumption and actual running time (on GPU) and further proposes a ViT design that performs favourably in those metrics while providing high accuracy when used as a backbone in various vision tasks, namely: image classification, object detection, instance segmentation and semantic segmentation.

The proposed ViT architecture is based on separating the MultiHead Self Attention (MSA) heads into 2 groups - one group performs local window self attention to capture local fine grained details characterised by high frequencies and the second group performs global self attention on a downscaled (in practice - average pooling in each high res window) version of the feature map to capture global structures characterised by low frequencies. The total number of MSA heads are divided between the groups such that 1-alpha of the heads belong to the first group (local windowed self attention on the full resolution feature map) and alpha of the heads belong to the second group (global attention on the downscaled feature map). Their method is thus dubbed HiLo to denote the different attention branches working on High and Low frequencies. Regarding the value of alpha - the authors provide an experiment to measuring the effect of different choices of alpha and when measuring on the various benchmarks alpha is set to 0.9, so in practice 10% and 90% of the MSA heads belong to the high and low frequencies branch, respectively. Note that in the low frequency brach, keys and values are computed on the downscaled feature map, but the queries still come from the high frequency branch. Also, to further speed up the method, the authors replace explicit positional encoding by adding a layer of 3x3 depth-wise convolution in each Feed Forward block.

Finally, to demonstrate the effectiveness of their approach, the authors compare their methods to other ViT architecture in classification on ImageNet 1K as well as when using their architecture as a backbone (weights are initialised from the ImageNet trained model) in object detection and instance segmentation (measured on COCO) and semantic segmentation (measured on ADE20K). The experiments demonstrate relatively high speed, low number of FLOPS and high accuracy of the proposed method compared to other ViT architectures and efficient attention mechanisms.


**Questions:**

1. Suggestion - I would suggest to compare the proposed method against more recent ViT architectures.
2. Question - In the low frequency branch, why do the queries (Q matrix) still computed from the high resolution feature map? did you try to completely separate the two branches such that the queries will also be computed from the pooled feature map, in addition to K and V?

**Limitations:**

The authors have adequately addressed the limitations and potential negative societal impact of their work.

**Strengths And Weaknesses:**

Strength:
1. The paper highlights an important limitation of previous methods - that using previously proposed metrics such as memory footprint and number of FLOPs (and theoretic asymptotic complexity) are only proxies to the metric of running time (or throughput) and in practice methods that perform favourably in those metrics (FLOPs, memory) might actually have slow running time due to not being "hardware friendly". To the best of my knowledge, very few papers report throughput (an exception is the Swin paper) and addressing this limitation by providing an architecture design with"hardware friendliness" in mind is an important contribution of the paper.
2. The proposed method is simple and tackles efficient ViT design from a viewpoint that has not been considered before. The architecture separates handling of local and global information is simple, yet novel and sound.
3. The paper is well written and easy to follow.
4. The paper provide extensive experiments and visualisations.

Weaknesses:
1. In terms of technical contribution - while the proposed architecture is new, all the ideas presented were previously explored. For example, computing local windowed self attention have been previously used in Swin and Focal transformer (in different ways). Computing global self attention on pooled versions of the feature map have been previously explored in PVT and Focal transformer (this is also true for the removal of positional encoding and replacing with depth-wise convolutions). The paper does combine those ideas in a way that was not previously suggested, but in my opinion this doesn't suffice for a Neurips paper (in terms of technical novelty).
2. Regarding the experiments - the methods compared against the proposed architecture are relatively old and are missing several strong recent papers, for example: XCiT, CaiT and CoAtNet to name a few. While HiLo performs favourably agains older methods, more recent works (as mentioned above) report higher accuracy than HiLo. Of course, their speed might be lower, but this needs to be tested and compared against HiLo to support the authors claims.

**Edit**: The rebuttal addresses my main concerns and the weaknesses I described above. I accept the authors claims that the proposed design principals have not been considered before and specially IMO designing an efficient architecture with high throughput is a significant contribution. Also, the experiments do demonstrate that the proposed method is on par with previous contributions (in my comment I was addressing performance when fine-tuning on 324x324) while being more efficient. I have changed the final rating accordingly, I would like to thank the authors again for the detailed rebuttal.

---

> ### Author Response · Authors · 2022-08-02
> **Response to Reviewer xTew**
>
>
> Thanks for taking the time to review our paper and we address your questions as follows.
>
> **Q1. Questions about the technical contribution.**
>
> **1)** As discussed in Lines 90-98 (Lines 90-96 of the initial submission), existing attention mechanisms in ViTs suffer from different problems, e.g. lacking either local or global attention, or having slow throughput on GPUs. Therefore, designing a fast attention mechanism that simultaneously captures local and global dependencies for ViTs is non-trivial. To this end, the main novelty of this paper is **a new efficient co-design of local and global self-attention, along with a novel fast ViT backbone**, as recognized by Reviewer VgYF, rather than improving local window attention or global attention separately. Besides, as mentioned in the strength, we also provide an architecture design with "hardware friendliness" in mind,  which “is an important contribution of the paper”.
> **2)** Furthermore, the motivation and design of HiLo come from a novel perspective where it efficiently disentangles low/high frequencies in a self-attention layer (Lines 48-54). We have conducted comprehensive benchmarking and shown that HiLo outperforms representative attention mechanisms on both the ImageNet pretraining and the downstream semantic segmentation task (Table 4 in the revised submission).
>
>
> **Q2. Compared with more ViT architectures.**
>
> Thanks for your advice. We agree that there are many ViTs in the literature. However, we would like to point out that XCiT, CaiT, and CoAtNet do not achieve better performance than ours **under a similar model size and the same experimental setting (i.e. image resolution of 224 $\times$ 224)**. For example, as shown in the table below, LITv2 performs favorably against others while achieving faster speed and lower model complexity. Note that we also believe further improvement can be achieved via model scaling, pretraining on larger datasets (e.g. ImageNet-22K), and finetuning on higher image resolution (e.g. 384, 512). However, such experiments are computationally expensive which is beyond our current hardware capacity. At this stage, we have adopted fair and standard experimental settings and compared LITv2 with many recent strong ViTs across different model sizes and tasks. In this case, we believe our experiments are quite comprehensive, which is recognized by Reviewer zE7q and Reviewer VgYF. We have added comparisons with more ViTs in the revised supplementary file.
>
>
> | Model       | Params (M) | FLOPs (G) | Throughput (images/s) | Top-1 (%) |
> | ----------- | ---------- | --------- | --------------------- | --------- |
> | XCiT-S12    | 26         | 4.8       | 1,068                 | 82.0      |
> | CaiT-XS24   | 27         | 5.4       | 623                   | 81.8      |
> | CoAtNet-0   | 25         | 4.2       | 1,151                 | 81.6      |
> | **LITv2-S** | 28         | **3.7**   | **1,471**             | **82.0**  |
> | XCiT-S24    | 48         | 9.1       | 612                   | 82.6      |
> | CaiT-S24    | 47         | 9.4       | 454                   | 82.7      |
> | CoAtNet-1   | 42         | 8.4       | 582                   | 83.3      |
> | **LITv2-M** | 49         | **7.5**   | **812**               | **83.3**  |
>
>
>
> **Q3. Directly computing queries from pooled feature maps.**
>
> In self-attention, the number of queries determines the spatial size of the output feature maps. When computing the queries from pooled feature maps, the spatial size of the output feature maps is inconsistent with that of the original input. One solution is to use interpolation (e.g. bilinear) and concatenate the interpolated feature maps with the outputs from Hi-Fi. However, as shown in the table below, this approach (denoted as "pooled queries") brings inferior performance and much slower throughput than our proposed design. Note that although computing queries from pooled feature maps can slightly achieve a lower theoretical model complexity, frequently applying interpolation on GPUs results in a high memory access cost (MAC). Therefore, it instead slows down the inference speed on GPUs. We have added this ablation study in the revised supplementary file.
>
> | Model                   | Params (M) | FLOPs (G) | Throughput (images/s) | Top-1 (%) |
> | ----------------------- | ---------- | --------- | --------------------- | --------- |
> | LITv2-S, this paper     | 28         | 3.7       | **1,471**             | **82.0**  |
> | LITv2-S, pooled queries | 28         | 3.5       | 1,084                 | 81.9      |

---

> > ### Comment · Reviewer_xTew · 2022-08-08
> > **Response to authors rebuttal**
> >
> > Dear authors,
> >
> > Thank you for your comments, added experiments (table IV) and explanations. You have addressed my main concerns as I wrote in detail in the edit of the official review.
> >
> > Best regards,
> >
> > Reviewer xTew

---

> > > ### Author Response · Authors · 2022-08-09
> > > **Thanks for your effort and suggestions!**
> > >
> > > Thanks for your valuable feedback and suggestions! We are glad to address your questions and appreciate your constructive reviews for improving our work.

---

> ### Author Response · Authors · 2022-08-07
> **Request for discussion**
>
> Dear Reviewer xTew,
>
> We sincerely thank you again for your great efforts in reviewing this paper. We have addressed your main concerns regarding our technical novelty and more comparisons with recent ViTs. Please don’t hesitate to let us know if there are still some concerns/questions.
>
> Best regards,
>
> Authors of #310

---

### Official Review · Reviewer_zE7q · 2022-07-12

**Rating:** 5
**Confidence:** 5
**Soundness:** 3 good
**Presentation:** 3 good
**Contribution:** 2 fair

**Summary:**

This paper studies the efficient Vision Transformers. The authors proposed the HiLo attention which combines the window-based self-attention and spatial reduction self-attention. The experimental results on serval datasets demonstrate the effectiveness of the proposed method.


**Questions:**

Please refer to the Questions in the Weaknesses.

How do you design the number of multi-head in attention when splitting the channels? Because the split features may not be divisible by the head_dim.


**Limitations:**

Yes.

**Strengths And Weaknesses:**

+ The proposed HiLo attention is somehow reasonable.
+ Compared to the baselines, the proposed methods could bring constant improvements
+ The abundant experiments are introduced to prove the effectiveness of the proposed method.

The main concerns are listed below.
- The core contribution is HiLo attention, however, as shown in Table 5, Compared to adding ConvFFN, the improvements HiLo attention brings are minor.
- In Figure 4, the HiLo attention is equivalent to spatial reduction attention (SRA) when the alpha=1.0. SRA could achieve ~81.9 Top-1 Accuracy on ImageNet with 3.7G FLOPs. Table 4 shows that HiLo (alpha=0.9) attention achieves 82.0 with 3.7G FLOPs. The improvement is just 0.1%, which is too weak.
- some references are missing.
 For window-based self-attention, there are some methods [a][b][c] are not included.
 To study high/low frequencies in images, Octave Convolution[d] is heavily related to the proposed method.

[a] MixFormer: Mixing Features across Windows and Dimensions, CVPR 2022.

[b] Transformer in Transformer, NeurIPS 2021.

[c] Rethinking Spatial Shuffle for Vision Transformer, arXiv 2021.

[d] Drop an Octave: Reducing Spatial Redundancy in Convolutional Neural Networks with Octave Convolution, ICCV 2019.

---

> ### Author Response · Authors · 2022-08-02
> **Response to Reviewer zE7q - Part 1**
>
> Thanks for your constructive comments and we address your questions as follows.
>
> **Q1. Compared to adding ConvFFN, the performance improvements HiLo attention brings are minor.**
>
> We would like to point out that ConvFFN and HiLo are proposed to address different bottlenecks in the proposed architecture, as described in Lines 133-138 (Lines 130-135 in the initial submission). Specifically, ConvFFN improves the **efficiency of positional encoding** in LITv1 while simultaneously enlarging the early receptive fields, thus improving the performance. On the other hand, HiLo mainly focuses on improving the **efficiency of self-attention**, especially when handling high-resolution dense prediction tasks, and thus it brings a better speed and accuracy trade-off. For example, in Table 5, we have shown that HiLo helps to reduce 27% FLOPs and achieve a $1.4\times$ speedup on COCO object detection. Moreover, in our response to Q2 below we show that HiLo achieves more performance gain than other attention mechanisms on semantic segmentation.
>
> **Q2. HiLo brings weak improvement over SRA on ImageNet-1K.**
>
> It is worth noting that HiLo is different from SRA in both the motivation and the concrete approach, even with alpha = 1.0. Specifically, SRA applies **conv-layernorm** to reduce the spatial size of keys and values for complexity reduction, which introduces more model parameters and neglects the importance of local fine details as well as the different frequencies in natural images. Compared to it, HiLo is motivated from a novel perspective on the frequency domain, as clearly indicated in Lines 48-54. Moreover, the Lo-Fi branch in HiLo applies **average-pooling** in order to obtain the low-frequency signals in each window, which is parameter-free and complementary to the Hi-Fi branch which captures high-frequency information in each window.
>
> Furthermore, we agree that the pure Lo-Fi branch can achieve competitive results on ImageNet-1K. However, **we would like to point out that high-frequency signals play an important role in capturing fine object details, which is particularly important for dense prediction tasks such as semantic segmentation**. For example, we train LITv2-S with Semantic FPN under different attention mechanisms on ADE20K. As the table below shows, HiLo with alpha=0.9 achieves superior performance compared to SRA (+1.5%) as well as other attention mechanisms. It indicates that both high/low frequencies are important in CV tasks and the effect of high frequencies is more significant in dense prediction tasks. We have added this comparison in Section 5.4 in the revision.
>
>
>
> | Backbone Attention in LITv2-S | Params (M) | FLOPs (G) | mIoU (%)    |
> | ----------------------------- | ---------- | -------- | -------- |
> | MSA                           | 32         | 46.5     | 43.7     |
> | SRA (PVT)                     | 35         | 42.4     | 42.8     |
> | W-MSA (Swin)                  | 32         | 42.7     | 41.9     |
> | T-MSA (Twins)                 | 33         | 42.5     | 44.0     |
> | HiLo w/ Alpha = 1.0           | 32         | 42.5     | 43.7     |
> | HiLo  w/ Alpha = 0.9          | **31**     | 42.6     | **44.3** |

---

> ### Author Response · Authors · 2022-08-02
> **Response to Reviewer zE7q - Part 2**
>
> **Q3. Some references are missing.**
>
> Thanks for pointing it out. We have included the following discussions in Section 2 in the revision. Specifically,
>
> - MixFormer [a] mixes local window attention with depthwise convolution, while the proposed HiLo is a novel attention mechanism that simultaneously captures local and global dependencies without introducing additional model parameters.
> - TNT [b] relies on additional tokens in the architecture to achieve global interaction, while the proposed HiLo directly captures both local and global dependencies of the original feature map in a single self-attention layer as a drop-in-replacement module. We also show in our response to Reviewer Rjxy Q3 that LITv2-S beats TNT-S [b] in terms of faster throughput on GPUs and better performance.
> - Similar to shifted window attention (Swin) which depends on window shifting to mix tokens among different windows, Shuffle Transformer [c] applies token shuffling among windows. However, both methods focus on local attention at the same self-attention layer, unlike HiLo which simultaneously captures local and global dependencies.
> - Octave convolution [d] shares a similar motivation with HiLo, i.e. disentangling different frequencies in a feature map. However, we are different in both the underlying problem target and the concrete approach. Specifically, Octave convolution targets convolutional layers, which is a type of convolution that applies locally on high/low-resolution feature maps separately. On the other hand, HiLo aims to boost the efficiency of self-attention in ViTs, which is a novel attention mechanism that captures both local and global relationships with self-attention.
>
> **Q4. How to design the number of multi-head in attention when splitting the channels.**
>
> In our implementation, we use the hyperparameter of alpha to control the number of heads in Hi-Fi and Lo-Fi. For example, with `alpha = 0.9` and `num_heads = 12` , we allocate `round(12 * 0.9) = 10 ` heads for Lo-Fi and another `12 - round(12 * 0.9) = 2` heads for Hi-Fi. We have provided this code in our supplementary material (Lines 39-40 in supp_code/models/attentions.py).

---

### Meta-Review · Area_Chair_9JRw · 2022-08-26

**Recommendation:** Accept
**Confidence:** Certain

**Metareview:**

Initially, this paper received diverging reviews. The authors did a good job addressing the reviewers' concerns, by adding additional comparisons to more SOTA ViT backbones and benchmarking throughput on a variety of GPU platforms. The AC agrees with the reviewer that the concerns have been sufficiently addressed and recommends acceptance.

**Award:**

No

---

### Decision · Program_Chairs · 2022-09-14

Accept